# Prediction of Persistent Tumor Status in Nasopharyngeal Carcinoma Post-Radiotherapy-Related Treatment: A Machine Learning Approach

**DOI:** 10.3390/cancers17010096

**Published:** 2024-12-31

**Authors:** Hsien-Chun Tseng, Chao-Yu Shen, Pan-Fu Kao, Chun-Yi Chuang, Da-Yi Yan, Yi-Han Liao, Xuan-Ping Lu, Ting-Jung Sheu, Wei-Chih Shen

**Affiliations:** 1Department of Radiation Oncology, Chung Shan Medical University Hospital, Taichung 40201, Taiwan; radtseng48@gmail.com; 2School of Medicine, Chung Shan Medical University, Taichung 40201, Taiwan; shenchaoyu@gmail.com (C.-Y.S.); pfkao@yahoo.com.tw (P.-F.K.); cyi4602@gmail.com (C.-Y.C.); 3Department of Medical Imaging, Chung Shan Medical University Hospital, Taichung 40201, Taiwan; 4Department of Nuclear Medicine, Chung Shan Medical University Hospital, Taichung 40201, Taiwan; twhappy17@gmail.com; 5Department of Otolaryngology, Chung Shan Medical University Hospital, Taichung 40201, Taiwan; 6Artificial Intelligence Center, Chung Shan Medical University Hospital, Taichung 40201, Taiwan; kirito203203@gmail.com (D.-Y.Y.); complicatedwhsh@gmail.com (Y.-H.L.); tingjungsheu@gmail.com (T.-J.S.); 7Department of Medical Informatics, Chung Shan Medical University, Taichung 40201, Taiwan

**Keywords:** nasopharyngeal cancer, radiotherapy-related treatment, treatment response, persistent tumor status, FDG PET/CT, radiomics, prediction model, machine learning

## Abstract

Radiotherapy is the primary and only curative treatment for nasopharyngeal carcinoma (NPC). The adoption of intensity-modulated radiotherapy (IMRT) alone or combined with optimized chemotherapeutic strategies has improved survival rates while reducing treatment-related toxicities. However, the persistent tumor status, including residual tumor presence and early recurrence, is the predominant cause of treatment failure, associated with poorer survival outcomes. This study extracted the radiomic features from pretreatment PET images, which were used to construct a prediction model to identify patients with NPC at high risk of having persistent tumors after treatment. The model trained by the AdaBoost algorithm showed significant diagnostic performance in predicting treatment failure. This model exhibited the feasibility of assisting clinicians in identifying high-risk patients before treatment, allowing for more personalized treatment plans and improving patient outcomes.

## 1. Introduction

Nasopharyngeal carcinoma (NPC) is a form of head and neck cancer originating in the epithelial tissue of the nasopharynx. Relative to other malignancies, NPC is uncommon [1,2]. According to the International Agency for Research on Cancer, an estimated 120,416 new cases of NPC were diagnosed in 2022, with these diagnoses representing only 0.6% of all cancer diagnoses globally in that year [2]. NPC exhibits a distinct geographical distribution [1,2], with the highest incidence and mortality rates reported in East Asia, followed by in Southeast Asia and South–Central Asia. A consistent male predominance in both incidence and mortality is observed across all regions worldwide.

Radiotherapy is the primary and only curative treatment for NPC [3]. The widespread adoption of intensity-modulated radiotherapy (IMRT) combined with optimized chemotherapeutic strategies, including induction, concurrent, and adjuvant strategies, has improved survival rates while reducing treatment-related toxicities [4]. However, despite these advances, 20–30% of survivors experience recurrent disease at the primary or local site, with this being a predominant cause of treatment failure [5,6]. The temporal risk of NPC recurrence sharply peaks at 24 months post-treatment, with 50–60% of recurrences occurring within this critical period [7,8,9,10]. Early recurrence is a key prognostic indicator for NPC that is strongly associated with poorer survival outcomes [4,7,8,9,10,11]. Managing recurrent NPC requires a multidisciplinary approach that incorporates personalized therapeutic strategies tailored to individual clinical profiles [12]. However, balancing the potential for salvage treatment with the risk of severe adverse effects related to reirradiation remains a considerable challenge [6]. Therefore, timely identification and intervention are crucial to improving management and prognosis for NPC. Notably, several reliable artificial intelligence applications have been developed that can assist with treatment decision-making, providing a promising avenue for enhancing outcomes in NPC.

The integration of radiomic features with machine learning has demonstrated considerable potential in predicting treatment response and survival outcomes for patients with NPC [13]. Studies have applied magnetic resonance imaging (MRI)-based radiomics before treatment to predict eligibility for adaptive radiation therapy [14], assess responses to neoadjuvants [15] or induction chemotherapy (ICT) [16], and compare responses across different chemotherapy regimens [17]. These radiomics have also been employed to investigate radioresistance through the analysis of recurrence patterns in patients treated with IMRT [18] and used in combination with machine learning to predict local and distant failure [19]. The predictive value of multiparametric MRI-based radiomics was also explored for predicting ICT response and survival outcomes [20]. Additionally, the prognostic system integrating MRI features and clinical data study was developed and validated to improve risk stratification and treatment decisions for locoregionally advanced NPC patients receiving concurrent chemoradiotherapy (CCRT) [21]. PET/computed tomography (CT)-based radiomics combined with machine learning was used to differentiate between local recurrence and inflammation in post-treatment patients [22]. Deep learning–based contrast CT and PET radiomics were reported to have capabilities for risk stratification, personalized ICT planning, and predicting disease-free survival [23]. Furthermore, a study employed positron emission tomography/computed tomography (PET/CT)-based deep learning signatures to stratify patients with recurrent NPC into high- and low-risk groups to enhance survival outcome predictions [24]. Radiomics and deep learning models [25,26,27,28] have also been applied extensively in this context to further improve prediction accuracy. The nomogram based on CT radiomics and clinical factors established the risk prediction of local recurrence in NPC patients after IMRT [29]. Research indicates that tumor response can be predicted by detecting anatomical changes in NPC during treatment [30]. Furthermore, a systematic review and meta-analysis evaluated the prognostic value of MRI-based radiomics in predicting progression-free survival in patients with NPC. Although the findings indicated robust prognostic performance, they highlighted a need for consistent and robust study protocols to strengthen reliability and clinical applicability [31].

In this study, we introduce a machine learning approach to predict persistent tumor status in NPC patients undergoing radiotherapy-related (RT-related) treatments. This approach leverages radiomic features extracted from pretreatment PET images to construct an explainable prediction model. The patient cohort is not limited to a specific stage but includes all patients receiving radiotherapy, whether as a standalone treatment or combined with various chemotherapy plans. The innovations of this study include capturing the biological characteristics of tumors through radiomic features, applying machine learning techniques to construct a model for predicting treatment response and analyzing the predictions to explore the correlation between radiomic features and treatment outcomes. Our contributions to the literature include providing a new predictive tool for clinicians to assess therapeutic effects and potentially improve personalized treatment strategies for NPC patients.

The structure of this paper is organized as follows: the Section 1 provides background information on NPC and the importance of the early prediction of treatment response. The Section 2 details the study design, patient population, patient treatment and follow-up methods, imaging processing techniques, and machine learning algorithms used to develop the prediction model. The Section 3 presents the diagnostic performance of radiomic features and the model and analyzes the impact of radiomic features on predictions. The Section 4 interprets the findings, highlights the clinical implications, and addresses the limitations of the study. Finally, the Section 5 summarizes the key contributions and suggests directions for future research.

## 2. Materials and Methods

The protocol of this retrospective study was approved by the Institutional Review Board of Chung Shan Medical University Hospital, and the requirement to obtain informed consent was waived (no. CS2-22194). Additionally, this article was self-assessed using the CheckList for EvaluAtion of Radiomics research (CLEAR) [32] (Appendix A).

### 2.1. Study Design and Patient Population

The study design, involving the clinical evaluation process and the construction of a predictive model, is illustrated in Figure 1. This study retrospectively collected clinical data from all patients given a diagnosis of NPC at our hospital. The diagnostic evaluation tools included MRI, PET/CT, and Epstein–Barr virus (EBV) tests, with final diagnoses confirmed through biopsy. Patients were eligible for inclusion if they underwent RT-related treatments and pretreatment clinical staging with PET/CT and if complete follow-up data were available for at least 3 years post-treatment. Patients were classified as “disease-free” if no residual or recurrent tumor was detected during the follow-up period; otherwise, they were categorized as having “persistent tumor status”.

Pretreatment PET/CT images were annotated for each patient by using the maximum standardized uptake value (SUV_max_). Two metabolic tumor volumes, metabolic tumor volume (MTV)_50_ and MTV_60_, were delineated on the basis of thresholds calculated from SUV_max_ * 50% and SUV_max_ * 60%, respectively. From these volumes, the radiomic features (RFs) RF_50_ and RF_60_ and differential radiomic features, that is, RF_Diff_, were derived. A predictive model was constructed using various machine learning methods to investigate the relationship between radiomic features and persistent tumor status. The performance of each predictive model was evaluated through k-fold cross-validation to validate the robustness and to aggregate the validation results across folds to identify the optimal model. This cross-validation approach provides insights into the model’s performance on unseen data and demonstrates its generalizability. Finally, the radiomic features with the most notable predictive value were identified to elucidate the decision-making process of the most effective model.

### 2.2. Patient Inclusion Criteria

Figure 2 presents a flowchart detailing the patient selection process. Between January 2012 and February 2022, 220 patients with NPC received RT-related treatments at our hospital. A two-stage screening process was employed to select eligible participants for the study.

In the first stage, 86 patients were excluded because of the following reasons: thirty-eight patients either did not undergo a PET/CT examination for clinical staging or had damaged PET/CT image backups. Thirty-four patients had a history of NPC, were given a diagnosis of NPC and synchronous cancer, or had previously undergone local treatment for head and neck cancer or systemic treatment for other cancers. Additionally, 12 patients underwent therapies that did not conform to the 6 standard treatment protocols defined in this study. Furthermore, two patients were lost to follow-up. At the end of this stage, 134 patients with complete diagnostic and treatment data were deemed eligible for inclusion in further screening.

In the second stage, an additional 30 patients were excluded for the following reasons: nineteen disease-free patients had follow-up periods of <3 years. Seven patients experienced over-segmentation during the delineation of MTVs. Four disease-free patients died from causes unrelated to cancer during the follow-up period. Ultimately, 104 patients who met all inclusion criteria were included in this study.

### 2.3. Pretreatment PET/CT for Clinical Staging

All PET/CT examinations were performed using either a GE Discovery MI PET/CT scanner (GE Healthcare, Milwaukee, WI, USA) or a Philips Gemini GXL PET/CT scanner (Philips Healthcare, Cleveland, OH, USA). Prior to the examination, the patients were instructed to fast for at least 6 h, and their blood glucose levels were examined to ensure that they were <200 mg/dL. Approximately 60 min after an intravenous injection of 175–370 MBq (5–10 mCi) of fluorodeoxyglucose (FDG), with the amount determined by the patient’s body weight, a PET/CT scan was conducted. The scan covered the area from the vertex of the skull to the mid-thigh, in accordance with established guidelines for tumor imaging with FDG PET/CT [33]. CT images were obtained without the use of oral or intravenous contrast agents. Each CT scan began with a scout view at 10 mA and 120 kVp, followed by a helical CT scan with a 0.5 s rotation time at 120 kVp and 15–180 mA (auto mA), with a 3.5 mm section thickness. Upon completion of the CT imaging, the scanner was returned to the landmark position and the FDG PET scan was acquired in three-dimensional acquisition mode at 2 min per bed position. The FDG PET/CT images were reconstructed using a standard iterative algorithm (ordered-subset expectation maximization), and the CT data were used for attenuation correction.

### 2.4. RT-Related Treatment

Patients received RT delivered using either a 6 or a 10 MV linear accelerator by using the IMRT or volumetric modulated arc therapy technique. The treatment regimen comprised conventional fractionated RT involving doses of 1.8–2 Gy per fraction, with five daily fractions per week. All patients were treated in a supine position. For primary tumor management, a dose of 45–50 Gy was administered, followed by a tumor boost with reduced fields. The planned total dose for the primary tumor was 70–74 Gy, whereas the dose for positive neck lymph nodes ranged from 60 to 70 Gy. In addition to RT, concurrent cisplatin chemotherapy was administered at a dose of 30 mg/m^2^ weekly, starting from week 1 and continuing for 6 consecutive weeks during the course of RT.

### 2.5. Follow-Up

Post-treatment follow-up protocols for patients with NPC included MRI scans conducted at 3-month intervals during the first 2 years post-treatment, with the follow-up intervals extending to 6 months thereafter. Plasma EBV DNA tests were conducted every 3 months, and PET/CT scans were performed annually. Further biopsies were arranged when necessary. At 1 month after completion of RT, high-risk patients with NPC were prescribed oral UFT (tegafur–uracil) for 1 year. Disease-free status was defined as the absence of positive findings from any of the four follow-up tools (i.e., MRI, PET/CT, EBV DNA test, and biopsy) during the follow-up period. Conversely, the presence of positive findings from any of these tools during follow-up was considered to indicate persistent tumor status.

### 2.6. Extraction of Radiomic Features and Differential Radiomic Features

Image analysis was conducted using in-house developed software, with radiomic features calculated using open-source Pyradiomics [34]. The computing environment had the following specifications: an Intel Core i9 processor, 96 GB of RAM, and an NVIDIA GeForce RTX 3090 GPU. The software used included Python 3.9.12, numpy 1.26.4, pandas 1.5.2, scikit-learn 1.1.1, scipy 1.11.4, lightgbm 4.3.0, xgboost 2.0.3, and shap 0.46.0. Each patient’s pretreatment PET/CT image was adopted for analysis. The process began with the user positioning a reference point within the NPC on the PET/CT images. This reference point was considered the center of a volume of interest (VOI), with dimensions of 7 cm in width, height, and depth. A computer algorithm was then employed to automatically detect all local maxima of SUVs within the VOI. The user then identified the local maximum within the spatial extent of the NPC, corresponding to the SUV_max_.

On the basis of the SUV_max_, two thresholds were defined, namely SUV_max_ × 0.5 and SUV_max_ × 0.6. A computerized region-growing method was employed, and these thresholds generated two MTVs, MTV_50_ and MTV_60_. The radiomic features were categorized into three distinct groups. For the first category, first-order features, descriptive statistics were used to characterize the distribution of SUV values within the MTV. Our data analysis led to the classification of conventional PET features commonly used in clinical practice as a separate subgroup. This distinction enables a comparison between traditional clinically validated features and radiomic features. The second category, shape features, encompassed metrics that describe the morphological characteristics of the MTV. That is, 5 conventional PET features, 16 first-order features, and 15 shape features were calculated for each MTV.

SUVs within the MTV were quantified into discrete SUV levels, with the number of discrete SUV levels being 32. A set of probability-based features was calculated based on the probability of occurrence of each discretized SUV, and these features were classified into the first category. The spatial arrangement of the discretized SUVs was described using texture features, which were assigned to the third category. Five matrices were used to depict this arrangement, namely the gray-level (GL) co-occurrence matrix (GLCM), GL dependence matrix (GLDM), GL run length matrix (GLRLM), GL size zone matrix (GLSZM), and neighboring gray-tone difference matrix (NGTDM). The definitions of these texture features were derived from GL image analysis. When applied to PET images, the discretized metabolic uptake (MU) is treated as analogous to the GL. For clarity, we used MU instead of GL in subsequent analyses. Thus, the five matrices were renamed as MUCM, MUDM, MURLM, MUSZM, and NMUDM. Specifically, 24, 14, 16, 16, and 5 features were extracted from MUCM, MUDM, MURLM, MUSZM, and NMUDM, respectively. That is, for each MTV, a total of 111 radiomic features were calculated. Finally, each differential radiomic feature was defined as the absolute value of the difference between corresponding features calculated from MTV_50_ and MTV_60_, yielding a total of 332 features. Because SUV_max_ is identical for both MTVs, no differential radiomic features were derived for SUV_max_.

### 2.7. Predictive Modeling Using Machine Learning Techniques

To explore the relationship between radiomic features, differential radiomic features, and post-treatment tumor status, we employed seven machine learning methods, which were AdaBoost, LightGBM, XGBoost, extra trees, random forest, logistic regression, and decision trees [35]. The objective was to develop a prediction model capable of accurately forecasting persistent tumor status in patients with NPC on the basis of these radiomic features.

For each machine learning model, recursive feature elimination (RFE) was adopted as a feature selection method to identify the radiomic features that contributed the most to the prediction. Initially, all features were included in the training of the prediction model, and the diagnostic performance was assessed as a baseline. RFE was then employed to recursively eliminate the least important features and rebuild the model with the remaining features. This process continued iteratively until the optimal subset of features was identified. RFE enhanced the model’s performance by eliminating redundant or less informative features, thereby enhancing computational efficiency and model interpretability.

The included patients were randomly divided into five independent datasets to train and validate the models developed using each machine learning method through five-fold cross-validation. In each round of cross-validation, four datasets (80% of the data) were used to train the machine learning model, whereas the remaining dataset (20% of the data) was used for validation. This process was repeated five times, with each dataset being used once as the validation set. After all five rounds were completed, the five validation results were aggregated to assess the overall predictive performance of each machine learning method. The predictive performance of the models developed using the seven machine learning methods was compared, and the method demonstrating the best performance was selected. The primary purpose of this step was to identify the best-performing algorithm by comparing the predictive performance of the models developed using the seven machine learning methods. The method demonstrating the best performance was then selected for further analysis.

SHapley Additive exPlanations (SHAP) values were employed to evaluate the importance of radiomic features by quantifying the contribution of each feature to the final model’s predictions [36]. Positive SHAP values indicate a bias toward predicting persistent tumor status, whereas negative SHAP values indicate a bias toward predicting disease-free status. The absolute SHAP values were ranked to assess the impact of individual features on the model’s predictive performance, which led to the identification of the most influential features for distinguishing between persistent tumor status and disease-free status. Analysis of the relationship between SHAP values and feature intensities provided valuable insights into the connection between radiomic features and post-treatment tumor status and highlighted biomarkers that could be clinically relevant for predicting treatment response in patients with NPC.

### 2.8. Statistical Analysis

The area under the receiver operating characteristic curve (AUC) was used to evaluate the predictive performances of radiomic features and the prediction models. Persistent tumor status was defined as a positive event in this study. An AUC value of >0.5 indicated that higher feature values were associated with an increased likelihood of persistent tumor status, whereas an AUC value of <0.5 indicated that higher feature values were more likely to correspond to disease-free status. Chi-square tests for independence and independent samples *t* tests were employed to assess the significant differences between clinical factors and post-treatment tumor status groups with a significance threshold set at *p* < 0.05. Survival analysis was performed using the Kaplan–Meier method. All statistical analyses were conducted using SPSS version 29.0 (IBM Corp, Armonk, NY, USA).

## 3. Results

### 3.1. Patient Characteristics

This study included 104 patients with NPC, with the patients divided into two groups on the basis of their post-treatment tumor status; 29 patients had persistent tumor status and 75 had disease-free status. The chi-square test revealed no significant differences in age (*p* = 0.830) and sex (*p* = 0.254) between the two groups (Table 1).

Histological and clinical staging revealed significant differences between the groups. Specifically, the difference in histological types (differentiated nonkeratinizing carcinoma vs. undifferentiated carcinoma) was significant (*p* = 0.037). Moreover, clinical distant metastasis stages (cM0 and cM1) were significantly associated with tumor status (*p* = 0.005), as was overall clinical staging (*p* = 0.036).

Regarding treatment modalities, no significant differences were observed in their impact on post-treatment tumor status. The primary treatment modalities (RT, CCRT, and ICT + CCRT) did not significantly influence post-treatment tumor status (*p* = 0.055). Additionally, the inclusion of maintenance therapy (*p* = 0.073) and the duration of maintenance therapy (≥1 year vs. <1 year, including those without maintenance therapy) did not significantly differ between the disease-free and persistent tumor status groups (*p* = 0.512).

### 3.2. Diagnostic Performance of Radiomic Features for Post-treatment Tumor Status

The SUV_max_, a conventional and widely used PET feature, had an AUC of 0.614 (±0.064), with a *p* value of 0.073, indicating weak diagnostic ability. Other conventional PET features, including MTV, TLG_max_, and TLG_mean_ calculated from MTV_50_ and MTV_60_, also exhibited nonsignificant diagnostic performance (*p* > 0.05). However, differential features of TLG_max_ and TLG_mean_ demonstrated significant potential, with *p* values of 0.036 and 0.023, respectively (Appendix A).

Within the first-order feature group, the differential features of energy, entropy, interquartile range, and total energy exhibited significant diagnostic performance (*p* = 0.002, 0.048, 0.036, and 0.002, respectively). In the shape feature group, differential features such as flatness, maximum 2D diameter row, and maximum 3D diameter exhibited a significant ability to distinguish post-treatment statuses (*p* = 0.026, 0.041, and 0.005, respectively).

Texture features derived from various matrices provided additional diagnostic insights. Significant diagnostic performance was observed for the maximal correlation coefficient in the MUCM, high dependence low MU emphasis in the MUDM, and large area low MU emphasis in the MUSZM, calculated from MTV_50_ and their corresponding differential features (all *p* < 0.05). Moreover, the differential features of MU nonuniformity normalized, low MU zone emphasis and small area low MU emphasis in the MUSZM exhibited significant diagnostic ability (*p* = 0.007, 0.035, and 0.004).

These findings highlight the diagnostic potential of radiomic features, particularly differential radiomic features, in effectively differentiating between disease-free and persistent tumor status.

### 3.3. Diagnostic Performance of AI Models in Predicting Persistent Tumor Status

Figure 3 presents the performance of the AI models in predicting persistent tumor status. The probabilities of persistent tumor status derived from the five validation sets were integrated and analyzed using receiver operating characteristic (ROC) analysis. Among the models, AdaBoost and LightGBM demonstrated superior predictive abilities, with AUC values of 0.934 and 0.896, respectively, although the difference did not reach significance (*p* = 0.43). However, LightGBM significantly outperformed the third-ranked model, XGBoost (AUC = 0.827, *p* = 0.004). Although AdaBoost and LightGBM had comparable performance, AdaBoost was selected as the final predictive model because of its higher AUC. ROC analysis was conducted to establish optimal thresholds for classifying patients into persistent tumor status and disease-free groups for each model. Table 2 presents the predictive performance of the seven models. The sensitivity, specificity, positive predicted value, negative predicted value, and accuracy of AdaBoost were 89.66%, 86.67%, 72.22%, 95.59%, and 87.5%, respectively.

### 3.4. Survival Analysis of Disease-Free and Persistent Tumor Status

Based on the AdaBoost model’s predictions, patients with NPC were classified into disease-free and persistent tumor status groups. As presented in Figure 4, the survival curves revealed a significant difference in overall survival between the two groups. Patients classified as disease-free demonstrated a markedly higher overall survival rate than did those with persistent tumor status. Moreover, the median overall survival for the disease-free group was significantly longer than that for the persistent tumor status group. Statistical analysis conducted using the log-rank test further confirmed that the difference in survival between the two groups was highly significant (*p* < 0.001).

### 3.5. Impact of Radiomic Features on Final Model in Predicting Persistent Tumor Status

The AdaBoost method was adopted to train the final predictive model by leveraging the radiomic features and hyperparameters identified during the cross-validation step and including the entire patient dataset. This approach ensured the model was trained using the maximum available data for enhanced interpretability. The objective of this step was not to further validate the model’s predictive performance but to gain insights into how the model interprets the data. A global interpretation of the impact of radiomic features and their mean absolute SHAP values are presented in Figure 5. Higher SHAP values indicated a more significant contribution of a feature to predicting persistent tumor status. The high dependence low MU emphasis_50_ exerted the greatest influence on the model’s prediction outcomes. Other notable features included surface area_50_, maximal correlation coefficient_Diff_, inverse variance_Diff_, and interquartile range_60_. These features ranked as the top five were pivotal in determining the likelihood of a tumor being classified as persistent.

Figure 6 plots the relationship between individual radiomic features and the prediction of post-treatment tumor status. Each dot represents a particular prediction, with the coordinate indicating feature values (*x*-axis) and the likelihood of persistent tumor status (*y*-axis). For the high dependence low MU emphasis_50_ feature, higher values were associated with a greater likelihood of predicting persistent tumor status (Figure 6a). A positive correlation was also noted for surface area_50_, with increasing values corresponding to a higher probability of persistent tumor status (Figure 6b). The third and fifth significant features, maximal correlation coefficient_Diff_ (Figure 6c), and interquartile range_60_ (Figure 6e) exhibited comparable patterns. Higher values for these features were also associated with an increased likelihood of persistent tumor status.

Inverse variance_Diff_ was the fourth most influential feature, and it exhibited complex patterns of influence on the model’s predictions (Figure 6d). Low values of inverse variance_Diff_ were distributed across both ends of the SHAP value spectrum, indicating that such values could increase the likelihood of predicting either persistent tumor status or disease-free status. Conversely, higher values of inverse variance_Diff_ were predominantly clustered around a SHAP value of approximately −0.01, indicating a slight tendency toward predicting disease-free status.

## 4. Discussion

This study demonstrated the potential of using radiomic features extracted from pretreatment PET images to effectively predict the treatment response in patients with NPC undergoing RT-related therapy, offering a promising avenue for personalized treatment strategies. The identified key features provided critical insights into the metabolic heterogeneity within the MTVs, shape characteristics of the MTVs, and texture differences between MTV_50_ and MTV_60_. These findings highlight the strong association between radiomic features and the biological properties of NPC, revealing their significant correlations with treatment outcomes. Moreover, the application of the AdaBoost algorithm enabled the construction of a robust predictive model by accurately capturing these complex relationships.

The interpretability of a prediction model is crucial to its clinical applicability. Therefore, SHAP analysis was conducted to identify the radiomic features that most contributed to the differentiation between persistent tumor status and disease-free status. The findings regarding the most influential feature, high dependence low MU emphasis_50_, indicated that regions with large, homogeneous, and low metabolic uptakes within MTV_50_ were strongly associated with a higher likelihood of persistent tumor status. The findings regarding the second most influential feature, surface area_50_, indicated that a larger surface area of MTV_50_ corresponded to an increased probability of persistent tumor status. The third key feature, maximal correlation coefficient_Diff_, which measures the complexity of texture, can be used to calculate the absolute difference between MTV_50_ and MTV_60_. A higher value for this feature was positively correlated with persistent tumor status. Interquartile range_60_ was the fifth most influential feature. The findings related to this feature indicated that a broader range of SUVs between the 25th and 75th percentiles within MTV_60_ was associated with a lower probability of being disease-free. Conversely, inverse variance_Diff_, which was the fourth most influential feature, is a measure of the difference in local homogeneity between MTV_50_ and MTV_60_.

These radiomic features effectively captured the biological characteristics of the tumor, serving as indicators of poor treatment response. Within the MTV, the presence of homogeneous, low-metabolism areas may indicate regions of hypoxia or necrosis [37,38]. Hypoxic tumor cells are often more resistant to radiation and chemotherapy because of their lower proliferation rates and altered metabolic pathways, which reduce their susceptibility to treatment-induced damage [39,40]. Additionally, a wide range of SUVs, a key indicator of high metabolic heterogeneity, indicates the presence of tumor regions with varying levels of glucose uptake and metabolic activity [37]. This metabolic heterogeneity is indicative of diverse tumor cell populations exhibiting distinct biological behaviors and treatment sensitivities. Furthermore, MTV with a larger surface area reflects a more complex and irregular geometry characteristic of a highly invasive tumor. Such tumors often have more substantial interactions with surrounding tissues. This invasiveness facilitates tumor spread and metastasis, leading to poorer responses to localized treatments.

Although FDG PET is a widely used tool for pretreatment staging, the complexity of glucose metabolism in cancer cells, influenced by the tumor microenvironment, poses challenges in terms of accurately depicting the heterogeneity of the tumor microenvironment according to FDG uptake levels and identifying prognostic factors. Therefore, radiotracers must be developed that target specific prognostic factors. For instance, hypoxia is a critical prognostic factor for the failure of RT and chemotherapy. Fluoromisonidazole (FMISO) is a specialized radiotracer designed to detect hypoxia [41]. However, despite holding considerable potential, such targeted tracers are less commonly used than FDG is. This may be because performing two PET scans with these tracers may lead to increased costs and higher patient radiation exposure. To address these challenges, researchers have explored the correlation between FDG and FMISO, and they have reported a moderate relationship between these tracers [41].

Our proposed predictive model can considerably enhance the clinical application of FDG PET for evaluating the treatment response of NPC. By identifying significant radiomic features related to metabolic activity and tumor volume shape, the model highlights the critical role of biological factors such as hypoxia and metabolic heterogeneity in influencing treatment outcomes. Furthermore, the treatment of locally advanced recurrent NPC remains a challenge because reirradiation after a high-dose RT course often results in severe late toxicity, potentially negating its therapeutic benefits. Emerging evidence suggests that hyperfractionated intensity-modulated RT can significantly reduce the incidence of such complications and improve overall survival rates [42]. By enabling the stratification of patients on the basis of predicted response, the model facilitates the development of personalized treatment plans tailored to each patient’s tumor characteristics. Being able to identify high-risk patients likely to respond poorly to standard treatment can allow clinicians to implement more intensive monitoring or consider more aggressive treatment options.

Despite its promising results, this study has several limitations that must be acknowledged. First, the study used patient data sourced from a single hospital, resulting in a relatively small sample size, which may limit the generalizability of the findings. To enhance the robustness and applicability of the predictive model, larger multicenter studies involving more diverse patient groups must be conducted. Second, the predictive model was developed using radiomic features extracted solely from pretreatment PET images. Future research should consider incorporating additional imaging modalities and clinical data to enhance the model’s accuracy. Addressing these limitations in future research is critical for ensuring that the predictive model can be effectively translated into routine clinical practice.

## 5. Conclusions

This study demonstrated the feasibility of using a machine learning approach to predict tumor persistence status in patients with NPC undergoing RT-related therapy. The significant radiomic features used to predict treatment response may reflect the biological characteristics of the tumor, including hypoxia, the heterogeneity of FDG uptake, and MTV shape. These findings suggest that radiomic features can serve as valuable biomarkers for predicting treatment outcomes in NPC patients. The prediction model developed in this study showed promising diagnostic performance, with high sensitivity, specificity, and accuracy. However, additional studies and clinical trials are required to validate this prediction model and to further explore the underlying biological mechanisms. Future research should also consider incorporating larger multicenter datasets and additional imaging modalities to enhance the model’s robustness and generalizability. Ultimately, integrating such predictive models into clinical practice could improve personalized treatment strategies and patient outcomes in NPC.

## Figures and Tables

**Figure 1 cancers-17-00096-f001:**
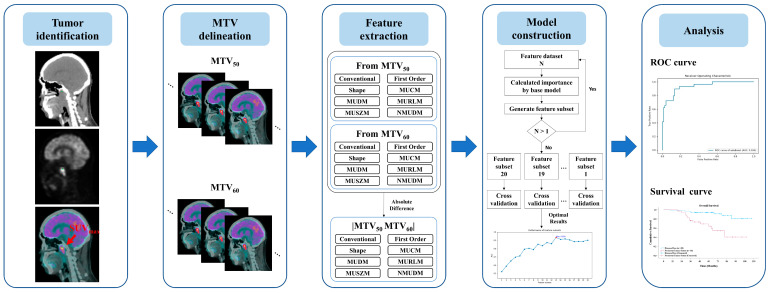
Overview of the study design illustrating the development of a machine learning model for predicting persistent tumor status in patients with NPC undergoing radiotherapy-related treatments.

**Figure 2 cancers-17-00096-f002:**
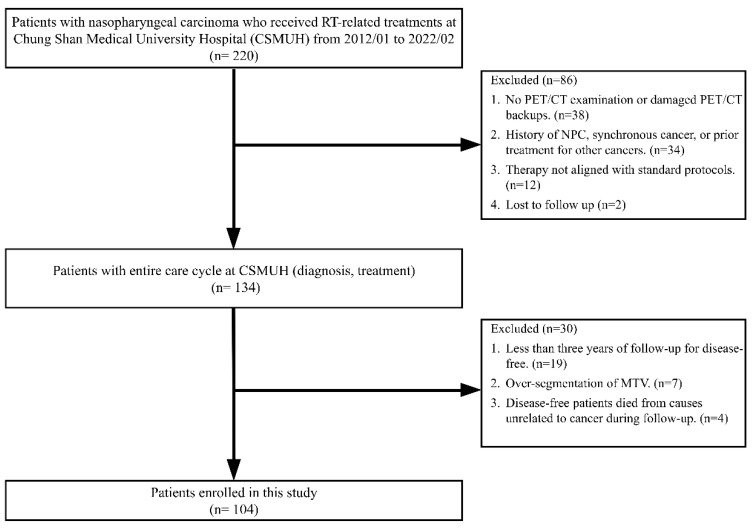
Flowchart of patient selection.

**Figure 3 cancers-17-00096-f003:**
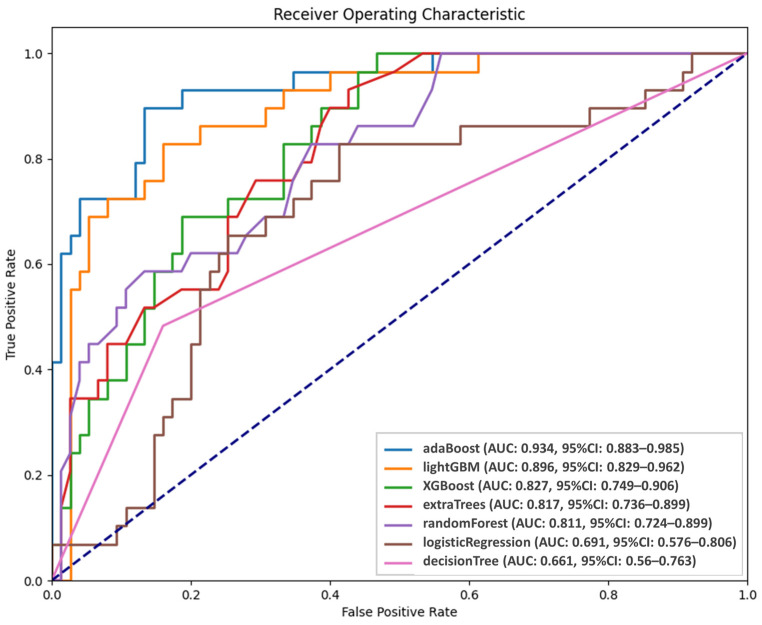
Diagnostic performance of AI models.

**Figure 4 cancers-17-00096-f004:**
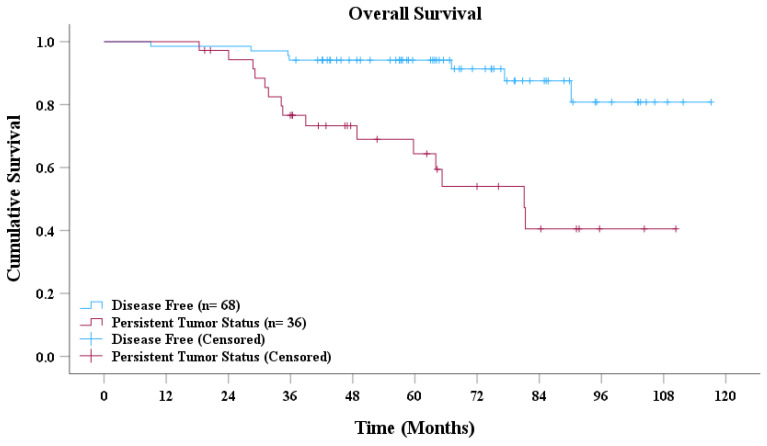
Overall survival curve.

**Figure 5 cancers-17-00096-f005:**
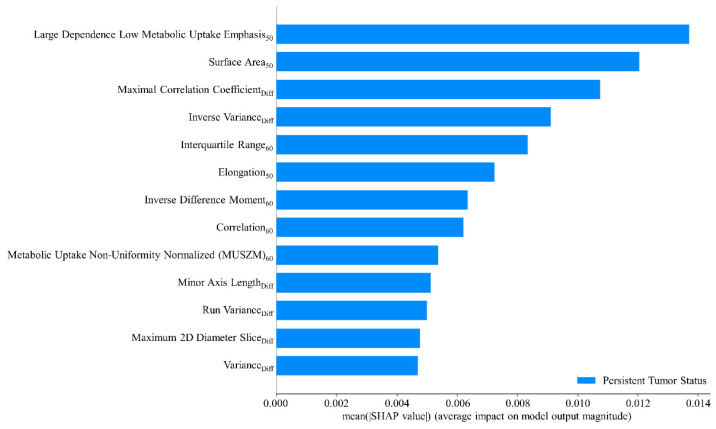
Mean SHAP value analysis of radiomic features.

**Figure 6 cancers-17-00096-f006:**
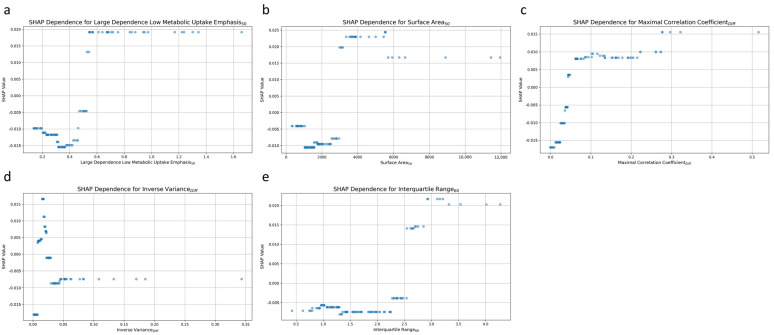
SHAP dependence plot for the top 5 radiomic features contributing to the AdaBoost model. (**a**) High dependence low metabolic uptake level emphasis_50_, (**b**) surface area_50_, (**c**) maximal correlation coefficient_Diff_, (**d**) inverse variance_Diff_, and (**e**) interquartile range_60._

**Table 1 cancers-17-00096-t001:** Patient characteristics (*n* = 104).

Clinical Factor		Post-Treatment Tumor Status		*p*-Value
		Disease-Free (n = 75)	Persistent Tumor Status (n = 29)	
Age (mean ± SD)		50.7 ± 11.4	51.2 ± 11.1	0.830
Sex				0.254
	Male	51 (68.0%)	23 (79.3%)	
	Female	24 (32.0%)	6 (20.7%)	
Histological type				0.037
	Differentiated carcinoma	5 (6.7%)	6 (20.7%)	
	Undifferentiated carcinoma	70 (93.3%)	23 (79.3%)	
Clinical distant metastasis stage				0.005
	cM0	71 (94.7%)	22 (75.9%)	
	cM1	4 (5.3%)	7 (24.1%)	
Clinical stage				0.036
	I	11 (14.7%)	1 (3.4%)	
	II	19 (25.3%)	2 (6.9%)	
	III	25 (33.3%)	14 (48.3%)	
	IV	20 (26.7%)	12 (41.4%)	
Treatment modalities				0.055
	RT	12 (16.0%)	1 (3.5%)	
	CCRT	27 (36.0%)	7 (24.1%)	
	ICT + CCRT	36 (48.0%)	21 (72.4%)	
RT-related treatment with MT				0.073
	RT alone	12 (16.0%)	1 (3.5%)	
	CCRT	9 (12.0%)	0 (0.0%)	
	CCRT + MT	18 (24.0%)	7 (24.1%)	
	ICT + CCRT	7 (9.3%)	5 (17.2%)	
	ICT + CCRT + MT	29 (38.7%)	16 (55.2%)	
Maintenance therapy (MT)				0.512
	≥1 years	39 (52.0%)	13 (48.3%)	
	<1 years or without MT	36 (48.0%)	16 (51.7%)	

**Table 2 cancers-17-00096-t002:** Diagnostic performance of constructed predictive models.

Predictive Model	Diagnostic Performance Indices				
	Sensitivity (95% CI)	Specificity (95% CI)	PPV (95% CI)	NPV (95% CI)	Accuracy (95% CI)
adaBoost	89.66% (78.57–100)	86.67% (78.97–94.36)	72.22% (57.59–86.85)	95.59% (90.71–100)	87.5% (81.14–93.86)
lightGBM	82.76% (69.01–96.51)	84% (75.7–92.3)	66.67% (51.27–82.07)	92.65% (86.44–98.85)	83.65% (76.55–90.76)
XGBoost	100% (100–100)	53.33% (42.04–64.62)	45.31% (33.12–57.51)	100% (100–100)	66.35% (57.26–75.43)
extraTrees	93.1% (83.88–100)	57.33% (46.14–68.53)	45.76% (33.05–58.48)	95.56% (89.53–100)	67.31% (58.29–76.32)
randomForest	82.76% (69.01–96.51)	62.67% (51.72–73.61)	46.15% (32.6–59.7)	90.38% (82.37–98.4)	68.27% (59.32–77.21)
Logistic Regression	82.76% (69.01–96.51)	58.67% (47.52–69.81)	43.64% (30.53–56.74)	89.8% (81.32–98.27)	65.38% (56.24–74.53)
Decision Tree	48.28% (30.09–66.46)	84% (75.7–92.3)	53.85% (34.68–73.01)	80.77% (72.02–89.52)	74.04% (65.61–82.46)

Abbreviations: PPV, positive predictive value; NPV, negative predictive value.

## Data Availability

The datasets generated during and/or analyzed in this study are available upon reasonable request from the corresponding author.

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
