# Peer review of "Prediction of Persistent Tumor Status in Nasopharyngeal Carcinoma Post-Radiotherapy-Related Treatment: A Machine Learning Approach"

_cancers, 2024, doi:10.3390/cancers17010096_

Round 1

Reviewer 1 Report

Comments and Suggestions for Authors

I have examined your study "Prediction of Persistent Tumor Status in Nasopharyngeal Carcinoma Post-Radiotherapy-Related Treatment: A Machine-Learning Approach" in detail. I have examined your study in detail. There are many spelling errors, including the abstract section. The abstract section meets the expectations. A paragraph should be added at the end of the introduction section, including the innovations of the article and its contributions to the literature. A paragraph should be added after this paragraph, including the article's organization. Figure 1 should be drawn more clearly. While detailed information is given about the number of patients in the article, no information is given about how many images of each patient were used. How were the training and test rates set in the classifiers? In what environment were the application results obtained? It is possible to support the results in the article with confusion matrices. In addition, how many features were extracted with each feature extraction method used? What was the path followed when these features were given to the classifiers? Were these features combined and given to the classifiers? No literature review was conducted on the subject. The conclusion section should be detailed.

Reviewer 2 Report

Comments and Suggestions for Authors

This is a useful addition to the radiomics literature. The study design, methods, results, and conclusions all appear to be reasonable and in line with what has come to be expected of work in this field.

My only minor recommendations for the revision are to add a few more specific details to various pieces of the methods and results:

1. Please include uncertainties (e.g., 95%CI) for the AUC curve in Figure 3, and for all reported numbers.

2. Although this study does not include a test set, if possible, some results demonstrating generalizability (e.g., out-of-bag error, or similar) would be informative.

3. Consider including a checklist for radiomics research (e.g., https://insightsimaging.springeropen.com/articles/10.1186/s13244-023-01415-8).

Reviewer 3 Report

Comments and Suggestions for Authors

The authors created a deep learning model, to evaluate the recurrence in HN case

The work is well done and clear also for me, that I know the concepts but I didn't do my myself this type of model.
They describe each steps and they justified what they did.

I have only a couple of commets

I explain my major issue
-    Page 7 Line 245: They wrote that divided the court in 5 group but in page 17 line 345 it is reported that they used a sub-group and after they applied the model to the full dataset. I know that the number of patient is not huge, they have to train the model on a sub group and after they have to apply it on "new patient" (non in the training or validation group) and show the result

Minor issue
-    The fig 6 has a bad quality and probably you can remove this session.

Round 2

Reviewer 1 Report

Comments and Suggestions for Authors

Congratulations on the successful revision.